# Efficient Weighting and Optimization in Federated Learning: A Primal-Dual Approach

## Abstract

Federated learning has emerged as a promising approach for constructing a large-scale cooperative learning system involving multiple clients without sharing their raw data. However, the task of finding the optimal sampling weights for each client, given a specific global objective, remains largely unexplored. This challenge becomes particularly pronounced when clients' data distributions are non-i.i.d. (independent and identically distributed) and when clients only partially participate in the learning process.

In this paper, we tackle this issue by formulating the aforementioned task as a bi-level optimization problem that incorporates the correlations among different clients. To address this problem, we propose a double-loop primal-dual-based algorithm, designed specifically to solve the bi-level optimization problem efficiently. To establish the effectiveness of our algorithm, we provide rigorous convergence analysis under mild assumptions. Furthermore, we conduct extensive empirical studies using both toy examples and learning models based on real datasets. Through these experiments, we verify and demonstrate the effectiveness of our proposed method.

## 1 Introduction

Federated learning has achieved high success in the large-scale cooperative learning system without sharing raw data. Its ability to operate on a massive scale, involving numerous devices, has garnered significant attention. However, with such a vast network of devices, ensuring the quality of data becomes an arduous task. After all, the presence of noisy or low-quality data can severely hinder a model's training effectiveness.

Recognizing this challenge, the question arises: how can we mitigate the influence of "bad" devices within the federated learning framework? A logical solution emerges - by reducing the weight assigned to these troublesome devices. Interestingly, popular federated training algorithms such as FedAvg (Li et al., 2019; He et al., 2020) often assign equal weights to all devices or base the weights on the number of data points they contribute. While these approaches have their merits, they fail to account for the varying quality of data provided by each device. Drawing inspiration from existing federated algorithms, we introduce a novel variable, denoted as $x$, which serves as a coefficient controlling the weight assigned to each device. By incorporating this weight control mechanism, we gain the flexibility to adjust the influence of individual devices on the model's learning process. To assess the impact of these coefficients on the model's performance, we introduce a validation set on the server. This validation set acts as a gauge, allowing us to determine if the coefficients improve the model's overall efficacy. We formulate the whole problem as a bi-level optimization as follows:

$$\min_{x} \quad f_0(w^*(x))$$

$$\text{s.t.} \quad w^*(x) \in \arg\min_{w} \sum_{i=1}^{N} x^{(i)} f_i(w) \tag{1}$$

$$x \in \mathcal{X} = \{x | x \geq 0, \|x\|_1 = 1\},$$

where $f_0$ is the validation loss evaluated in the global server and $f_i$ is the training loss evaluated on the $i_{th}$ client based on its own data. To solve problem 1, Kolstad & Lasdon (1990) propose an algorithm that calculates the gradient of $x$ directly, i.e.,

$$\frac{\partial f_0(w^*(x))}{\partial x^{(i)}} = -\nabla_w f_0(w^*(x))^\top \left( \sum_{i=1}^{N} \nabla_w^2 f_i(w^*(x)) \right)^{-1} \nabla_w f_i(w^*(x)).$$

However, due to the large parameter dimension of $w$, taking the inverse of the Hessian or solving the related linear system becomes computationally infeasible. Furthermore, estimating the gradient or Hessian of the local function $f_i$ directly is challenging due to the large amount of data on the local devices. Only stochastic gradient and stochastic Hessian can be accessed. In light of these constraints, Ghadimi & Wang (2018) proposed the FEDNEST algorithm, which approximates the inverse of the Hessian using a series of powers of the Hessian, represented by $\sum_{k=0}^{K}(I - \eta H)^k$ to approximate $\frac{1}{\eta} H^{-1}$ with a certain $\eta$. Similarly, Tarzanagh et al. (2022) introduced the FEDNEST algorithm for solving stochastic bi-level optimization problems in the federated learning setting. Both methods require an additional loop to approximate the product of the Hessian inverse with a vector.

However, it is known that for constraint optimization, with the descent direction, the algorithm will not converge to the optimal point or even to the first-order stationary point (Bertsekas, 2009). Therefore, obtaining an accurate approximation of the Hessian inverse becomes crucial. Because the series of powers must start with $k = 0$ and require several iterations to achieve accuracy, it increases computation and communication in federated learning. Fortunately, by leveraging the KKT condition, we can embed information about the Hessian inverse into dual variables. Based on the smoothness of the objective, we can initialize the dual variables effectively, instead of starting with the same initialization in each iteration (e.g., using $I$ in the series approximation). Consequently, we propose a primal-dual-based algorithm to solve the problem outlined in equation 1.

Furthermore, when solving constrained optimization problems with nonlinear equality constraints, adding the squared norm of the equality constraint as an augmented term may not introduce convexity to the augmented Lagrange function. Consequently, it becomes challenging for min-max optimization algorithms to find the stationary point. In contrast, following the assumption in Ghadimi & Wang (2018), where the functions $f_i$ are assumed to be strongly convex, adding the functions $f_i$ as the augmented term helps introduce convexity without altering the stationary point of the min-max problem. Based on this new augmented Lagrange function, we prove that by employing stochastic gradient descent and ascent, $w$ and $\lambda$ can converge to the KKT point. Additionally, using the implicit function theorem, when $w$ and $\lambda$ approach the stationary point of the min-max problem, the bias in estimating the gradient of $x$ can be reduced to 0. Thus, by combining the primal-dual algorithm on $w$ and $\lambda$ with stochastic projected gradient descent on $x$, we establish the convergence of our algorithm.

Finally, we conduct comparisons between our algorithm and other algorithms using both a toy example and real datasets, such as MNIST and F-MNIST with Network LeNet-5. The experimental results demonstrate that our proposed algorithm performs well in strongly convex cases and even exhibits effectiveness in some non-convex cases, such as neural networks. These findings provide compelling evidence of the capability and versatility of our algorithm in various scenarios.

We summarize our contributions as follows:

- In Federated Learning, we formulate the local coefficient learning problem as a bi-level optimization problem, which gives a way to identify the dataset quality in each local client for some specific task (where a small validation set is given).

- In bi-level optimization, we introduce a primal-dual framework and show the convergence of the whole algorithm in the constrained and stochastic setting.

- For some specific optimization problems with non-linear constraints, we give a new augmented term. With the new augmented term, the primal variable and dual variable can converge to the KKT point of the original problems.

## 2  Related work

### 2.1  Personalized Federated Learning

The most related work in federated learning tasks will be personalized federated learning. A well-trained local personalized model is needed for each local device in personalized federated learning. Jiang et al. (2019); Deng et al. (2020) propose a method that they train a global model and then fine-tune the trained global model to get the local model. T Dinh et al. (2020); Fallah et al. (2020) change the local objective function to make each local can be different and handle individual local tasks. Li et al. (2021) introduces a two-level optimization problem for seeking the best local model from great global models. All of these works do not involve a validation set as a reference, but they use a few gradient steps or simple modifications and hope the local model can both fit the local training data and use information from the global model (other local devices). Different from these works, we explicitly formulate a bi-level optimization problem. By adding a validation set, it can be more clearly identified the correlation between the other devices and its own.

### 2.2  Stochastic Bi-level Optimization

Bi-level optimization problem has been studied for a long time. One of the simplest cases in bi-level optimization is the singleton case, where the lower-level optimization has a unique global optimal point. Without calculating the inversion of the Hessian matrix of the lower-level optimization problem, there are two major algorithms. Franceschi et al. (2017) approximates $\frac{\partial w^*(x)}{\partial x}$ by $\frac{\partial w_T}{\partial x}$ where $w_T$ is the iterate after T steps gradient descent for the lower optimization problem. Using this method, in each iteration, we need to communicate $N$ (number of local devices) vectors among the server and local devices which is not communication efficient. The other method Ghadimi & Wang (2018) is to approximate $(\nabla_w^2 g(w))^{-1}$ by $\sum_{i=0}^{K}(I - \eta\nabla^2 g(w))^i$, where $g(w)$ is the objective function of lower-level optimization problem. Although Khanduri et al. (2021) points out that to approximate gradient for upper optimization function, we can get rid of taking the optimal point for lower optimization in each upper-level update optimization, which seems to get rid of double-loop approximation, it still needs a loop for approximating Hessian inverse with series. Guo & Yang (2021) uses SVRG to reduce the noise level of estimating stochastic gradient and Hessian to get better performance. Besides, all of the above works assume the smoothness of the local Hessian, but none of them will apply the property directly to the algorithm. Different from the above works, we introduce a primal-dual framework into bi-level optimization, where the dual variable can record the information of Hessian. Also, Shi et al. (2005); Hansen et al. (1992) introduce the primal-dual framework, but they stay in quadratic regime or mix integer programming, which is non-trivial to extend the results to federated learning settings. Further, Tarzanagh et al. (2022) introduce bi-level optimization into the federated learning setting.

Huang et al. (2022) propose to adaptively weight nodes, which is the most related work to ours. Different from Huang et al. (2022), we use a stochastic update for weight variable $x$, which makes us use a minibatch update for $x$ and partial participation for local clients.

## 3  Algorithm Design

Assume that each function of $f_i$ is a strongly convex function. Then, the optimal solution to the lower optimization problem becomes only a single point. Thus, with the implicit function theorem, we can calculate the gradient of $f_0(w^*(x))$ with respect to $x$ as follows.

**Proposition 1.** *Suppose $f_i$'s are strongly convex functions. Then for each $x \in \mathcal{X}$, it holds that $\frac{\partial f_0(w^*(x))}{\partial x^{(i)}} = -\nabla_w f_0(w^*(x))^\top \left( \sum_{j=1}^{N} x^{(j)} \nabla_w^2 f_j(w^*(x)) \right)^{-1} \nabla_w f_i(w^*(x)).$*

With the proposition 1, one can calculate the gradient of $x$, when $w^*(x)$ and the inverse of Hessian are given. However, for large-scale problems, none of these can be easily obtained. Fortunately, by noticing the convexity of each function $f_i$, we can replace the first constraint $w^*(x) \in \arg\min_w \sum_{i=1}^{N} x^{(i)} f_i(w)$ with

$\nabla \sum_{i=1}^{N} x^{(i)} f_i(w) = 0$. For given $x$, we can formulate the following constrained optimization problem:

$$\min_{w} \; f_0(w)$$
$$s.t. \; \sum_{i=1}^{N} x^{(i)} \nabla_w f_i(w) = 0, \tag{2}$$

By introducing the dual variable $\lambda$, we can easily get the Lagrange function. To solve the Lagrange function efficiently, we propose the following augmented Lagrange function.

$$L_x(w, \lambda) = f_0(w) + \lambda^\top \sum_{i=1}^{N} x^{(i)} \nabla_w f_i(w) + \Gamma \sum_{i=1}^{N} x^{(i)} f_i(w). \tag{3}$$

Different from the standard augmented terms, where the norm square of equality constraints is added to achieve strong convexity of the primal problem, we add the summation of $f_i$'s with coefficient $x^{(i)}$'s. If we use the norm square of the gradient constraint for general strongly convex functions, it will not be strongly convex. Thus, we can not directly adopt the gradient descent ascent algorithm. With the definition, we can obtain the following two propositions directly.

**Proposition 2.** *Suppose $f_i$'s are strongly convex functions for $i = 1, 2, \cdots, N$, $x^{(i)} \geq 0$ for all $i$ and $\|x\|_1 = 1$. Then, Problem 2 satisfies Linear Independence Constraint Qualification, and its KKT conditions can be written as follows:*

$$\nabla_w f_0(w) + \sum_{i=1}^{N} x^{(i)} \nabla_w^2 f_i(w) \lambda = 0$$

$$\sum_{i=1}^{N} x^{(i)} \nabla_w f_i(w) = 0.$$

**Proposition 3.** *Suppose $f_i$'s are strongly convex functions for $i = 1, 2, \cdots, N$, $x^{(i)} \geq 0$ for all $i$ and $\|x\|_1 = 1$. Then, the stationary point of $\min_w \max_\lambda L_x(w, \lambda)$ is unique and satisfies the KKT conditions of problem 2.*

Let $(\hat{w}^*(x), \lambda^*(x))$ be the stationary point of $\min_w \max_\lambda L_x(w, \lambda)$. From proposition 2, it holds that $\hat{w}^*(x) = w^*(x)$ and

$$\frac{\partial f_0(w^*(x))}{\partial x^{(i)}} = \lambda^*(x)^\top \nabla_w f_i(w^*(x)). \tag{4}$$

Thus, with the KKT point $w^*(x)$ and $\lambda^*(x)$, we can estimate the gradient of $x$ without estimating the inverse of Hessian. However, as $\lambda^\top \sum_{i=1}^{N} x^{(i)} \nabla_w f_i(w)$ can be a highly non-convex function, which can be harmful to the optimization process. We add an additional constraint on the norm of $\lambda$ and define the constraint set $\Lambda$. Thus, the problem 2 becomes

$$\min_{w} \max_{\lambda \in \Lambda} L_x(w, \lambda) = f_0(w) + \lambda^\top \sum_{i=1}^{N} x^{(i)} \nabla_w f_i(w) + \Gamma \sum_{i=1}^{N} x^{(i)} f_i(w). \tag{5}$$

We propose a double loop algorithm for solving problem 1. We show the algorithm in the Algorithm 1 and 2. In the inner loop, we solve the augmented Lagrange for $K$ steps. In each step, the local client will receive the iterates $w_{t,k}$ and $\lambda_{t,k}$. After that, each local client will calculate $\tilde{\nabla} f_i(w_{t,k})$ and $\hat{\nabla} f_i(w_{t,k})$ based on the backpropagation through two independent batches. The term $\tilde{\nabla}^2 f_i(w_{t,k}) \lambda_{t,k}$ is calculated with the auto-differentiable framework (i.e. Pytorch, TensorFlow) or with the closed-form multiplication. Then the local device sends gradient estimation $\tilde{\nabla}_w f_i(w_{t,k})$ and the estimated product of Hessian and $\lambda$ ($\tilde{\nabla}^2 f_i(w_{t,k}) \lambda_{t,k}$) to the server.

For the server, in each step, the server will first send the primal variable $(w_{t,k})$ and dual variable $(\lambda t, k)$ to all local clients. Then, the server will receive the estimated gradients and estimated products from some local clients. Because not all devices will stay online in each step, we define a set $Active_{t,k}$ which records the clients that participate in the optimization in the $(t, k)$ step. With the vectors collected from local clients, the server will calculate the gradient estimator of $w_{t,k}$ and $\lambda_{t,k}$ with respect to function $L_{x_t}(w_{t,k}, \lambda_{t,k})$. And then, $w_{t,k}$

---

**Algorithm 1** The bi-level primal-dual algorithm on local device $i$

---
1: **for** $t = 1, 2, \cdots, T$ **do**
2:     **for** $k = 1, 2, \cdots, K$ **do**
3:         Receive $w_{t,k}, \lambda_{t,k}$ from the server;
4:         Sample a mini-batch and calculate $\tilde{\nabla} f_i(w_{t,k})$;
5:         Sample a mini-batch and calculate $\hat{\nabla} f_i(w_{t,k})$;
6:         Calculate $\hat{\nabla}_w^2 f_i(w_{t,k})\lambda_{t,k}$ with back propagation on scalar $\hat{\nabla}_w f(w_{t,k})\lambda_{t,k}$;
7:         Send $\tilde{\nabla}^2 f_i(w_{t,k})\lambda_t$ and $\tilde{\nabla} f_i(w_{t,k})$ to the server;
8:     **end for**
9: **end for**

---

will be updated by a gradient descent step and $\lambda_{t,k}$ will be updated by a gradient ascent step. Different from local devices, after K inner loop update steps, based on the $\lambda_{t,K}$ and gradient estimated in each local client, the server will calculate the gradient of $x$ based on equation 4 and perform a projected gradient descent step on $x$. In addition, if the $i_{\text{th}}$ agent is not in $Active_{t,K}$, we set the gradient of $x^{(i)}$ to be zero.

---

**Algorithm 2** The Bi-level primal dual algorithm on the Server

---
1: **Initial** $x_1$, $w_{1,1}$, $\lambda_{1,1}$, **total iterations:** $K$, $T$ **and step size:** $\eta_w$, $\eta_\lambda$, $\eta_x$.
2: **for** $t = 1, 2, \cdots, T$ **do**
3:     **for** $k = 1, 2, \cdots, K$ **do**
4:         Send $w_{t,k}, \lambda_{t,k}$ to each local device;
5:         Receive $\tilde{\nabla} f_i(w_{t,k})$ and $\tilde{\nabla}_w^2 f_i(w_{t,k})\lambda_t$ from $Active_{t,k}$;
6:         $g_w = \tilde{\nabla} f_0(w_{t,k}) + \frac{N}{|Active_{t,k}|} \sum_{i \in Active_{t,k}} x_t^{(i)} \tilde{\nabla}_w^2 f_i(w_{t,k})\lambda_{t,k} + \Gamma \tilde{\nabla} f_i(w_{t,k})$;
7:         $w_{t,k+1} = w_{t,k} - \eta_w g_w$;
8:         $g_\lambda = \frac{N}{|Active_{t,k}|} \left( \sum_{i \in Active_{t,k}} x_t^{(i)} \tilde{\nabla}_w f_i(w_{t,k}) \right)$;
9:         $\lambda_{t,k+1} = \Pi_\Lambda \left( \lambda_{t,k} + \eta_\lambda g_\lambda \right)$;
10:     **end for**
11:     $g_{x^{(i)}} = \frac{N}{|Active_{t,K}|} \lambda_{t,K}^\top \tilde{\nabla}_w f_i(w_{t,K})$ for $i \in Active_{t,K}$;
12:     $g_{x^{(i)}} = 0$ for $i \notin Active_{t,K}$;
13:     $x_{t+1} = P_{\mathcal{X}}(x_t - \eta_x g_x)$;
14:     $\lambda_{t+1,1} = \lambda_{t,K+1}$;
15:     $w_{t+1,1} = w_{t,K+1}$
16: **end for**
17: **Output:** $x_T$, $W_{T,K+1}$.

---

**Remark 1.** *$g_{x^{(i)}}$ can be calculated in the i-th device and sent to the server, which can reduce the computation in the server and will increase one-round communication with one real number between the server and devices. The rest of the analysis will remain to be the same.*

## 4 Theoretical Analysis

In this section, we analyze the convergence property of the proposed algorithm. First, we state some assumptions used in the analysis.

**(A1)** $f_0, f_1, \cdots, f_N$ are lower bounded by $\underline{f}$, and $f_0, f_1, \cdots, f_N$ have $L_1$ Lipschitz gradient.

**(A2)** $f_1, \cdots, f_N$ are $\mu$-strongly convex functions.

**(A3)** $f_1, \cdots, f_N$ has $L_2$ Lipschitz Hessian.

**(A4)** $\max_{i \in \{0,1,\cdots,N\}} \max_{x \in \mathcal{X}} \|\nabla f_i(w^*(x))\| \leq D_w$.

**(A5)** Each local estimation is unbiased with bounded variance $\sigma^2$.

**(A6)** $Active_{t,k}$ *is independent and sampled from the set of the nonempty subset of* $\{1, 2, \cdots, N\}$*, where* $P(i \in Active_{t,k}) = p$ *for all* $i \in \{1, 2, \cdots, N\}$*.*

**Remark 2. (A1),(A2),(A3)** *are commonly used in the convergence analysis for bi-level optimization problems (Ji et al., 2021; Chen et al., 2021; Khanduri et al., 2021). Unlike Ji et al. (2021); Chen et al. (2021), where they need to assume* $f_0, f_1, \cdots, f_N$ *to be* $L_0$ *Lipschitz, we assume the gradient norm is bounded at the optimal solution. Because for machine learning models, regularization will be added to the objective function, making the norm of the optimal solution not large. When* $w^*(x)$ *can be bounded by some constant.* **(A4)** *is reasonable in practice. Moreover, the Lipschitz assumption on function can directly infer* **(A4)** *with* $D_w = L_0$*.* **(A5)** *is a common assumption used for stochastic gradient methods (Ghadimi et al., 2016) and* **(A6)** *extend the assumption in Karimireddy et al. (2020) by giving the probability that local devices will be chosen instead of uniformly sampling.*

**Remark 3.** *With* **(A4)***,* $D_\lambda = \max_{x \in \mathcal{X}} \|\lambda^*(x)\|$ *is upper bounded by* $D_w/\mu$*.*

**Proposition 4.** *When* $\Lambda = \{\lambda \mid \|\lambda\| \leq D_\lambda\}$*, then the stationary point of problem 5 is the KKT point of problem 2.*

With proposition 3 and 4, the stationary point of problem 5 is unique and we denote the stationary point as $(w^*(x), \lambda^*(x))$. To give the convergence of the whole algorithm, firstly, we give the convergence guarantee for the inner loop.

**Theorem 1.** *For given* $x \in \mathcal{X}$*, when (A1) to (A6) holds,* $\Gamma = \frac{D_\lambda L_2 + 2L_1}{\mu}$*,* $\eta_w \leq \frac{1}{L_L(1+p(1+\Gamma^2+D_\lambda^2))}$ *and* $\eta_\lambda = \frac{64(\Gamma\mu - D_\lambda L_2 - L_1)^2}{L_1^2(64 + (\Gamma\mu - D_\lambda L_2 - L_1)^2)}\eta_w \leq \frac{\mu^2}{4L_L(1+p)L_d^2}$*, when randomly choosing* $\hat{k} \in \{1, 2, \cdots, K\}$ *with equal probability it holds that*

$$\mathbb{E}\left[\left\|\lambda_{\hat{k}}^\top \nabla_w f_i(w_{\hat{k}}) - \frac{\partial f_0(w^*(x))}{\partial x^{(i)}}\right\|^2\right] \leq \frac{C_4}{K\eta_w} + C_5\eta_w + C_6\eta_w^2$$

*where*

$$C_4 = \frac{3(D_w^2 + D_\lambda L_1^2)(64 + L_1^2)\mu^4 + 12D_\lambda^2 L_L^2}{L_L^2}$$

$$C_5 = \left(\frac{L_L(\mu^2 + (d_\lambda L_2 + 2L_1)^2 + D_\lambda^2)}{2\mu^2} + \frac{32L_L^2 + \mu^2}{16L_L^2}\right)\sigma^2$$

$$C_6 = \frac{128L_1^2\sigma^2}{64 + L_1^2}$$

**Corollary 1.** *When selecting* $\eta_w = \Theta(1/\sqrt{K})$ *and* $\eta_\lambda = \Theta(1/\sqrt{K})$*, it holds that*

$$\mathbb{E}\left[\left\|\lambda_{\hat{k}}^\top \nabla_w f_i(w_{\hat{k}}) - \frac{\partial f_0(w^*(x))}{\partial x^{(i)}}\right\|^2\right] = O(1/\sqrt{K}).$$

Thus, with theorem 1, the gradient of $f_0$ with respect to $x$ can be "well" estimated through the inner gradient descent ascent method when the number of inner loop steps is large enough. Then, we can obtain the following convergence result of the outer loop.

**Theorem 2.** *Suppose (A1) to (A6) holds,* $\Gamma$*,* $\eta_w$ *and* $\eta_\lambda$ *are selected according to Theorem 1. We randomly choose* $\hat{k} \in \{1, 2, \cdots, K\}$ *with equal probability to approximate the gradient of* $x$*. Define* $\hat{x} = \arg\min_{y \in \mathcal{X}}(f_0(w^*(y)) + \frac{\rho}{2}\|y - x\|^2)$ *and* $\bar{\nabla}_\rho f_0(w^*(x)) = \rho(x - \hat{x})$ *for* $\rho = 2L_{f_0}$*, it holds that*

$$\frac{1}{T}\sum_{i=1}^{T}\sum_{t=1}^{T}\mathbb{E}\|\bar{\nabla}_\rho f_0(w^*(x_t))\|^2$$

$$\leq \frac{2L_{f_0}(f_0(w^*(x_1)) - \underline{f}) + 2L_{f_0}^2\|x_1 - \hat{x}_1\|^2}{T\eta_x} + 2L_{f_0}^2\delta^* + 4L_{f_0}^2((1+p)D_\lambda^2 D_w^2 + 2\delta^* + \sigma^2) + 2L_{f_0}\delta^*$$

*where* $L_{f_0} = \left(\frac{D_w^2}{\mu^2} + \frac{2D_w L_1}{\mu}\right)\frac{\sqrt{N}D_w\mu + \sqrt{N}D_w L_1}{\mu^3}$ *and* $\delta^* = \frac{C_4}{K\eta_w} + C_5\eta_w + C_6\eta_w^2$*.*

**Corollary 2.** *When we select $\eta_w = \Theta(1/\sqrt{K})$, $\eta_\lambda = \Theta(1/\sqrt{K})$ and $\eta_x = \Theta(1/\sqrt{T})$, it holds that*

$$\frac{1}{T}\sum_{t=1}^{T}\mathbb{E}\|\bar{\nabla}_\rho f_0(w^*(x_t))\|^2 = O(1/\sqrt{T} + 1/\sqrt{K}).$$

**Remark 4.** *To achieve $\epsilon$-stationary point ($\mathbb{E}\|\bar{\nabla}_\rho f_0(w^*(x_t))\|^2 \leq \epsilon$), $O(1/\epsilon^4)$ samples are needed in each local client and in the server. Different from the previous works on bilevel optimization(e.g. Ghadimi & Wang (2018), Khanduri et al. (2021) and Franceschi et al. (2017)), we prove the convergence when optimization variable x has a convex constraint.*

### 4.1 Proof Sketch of Theorem 1

To show the convergence of inner loop, we first construct a potential function for inner loop objective. Define $\Phi_x(w, \lambda) = L_x(w, \lambda) - 2d(\lambda)$, where $d(\lambda) = min_w L_x(w, \lambda)$ for given x. The intuition of defining this potential function is that $L_x(w, \lambda)$ is not necessarily decreasing in each iteration, as $\lambda$ is performing a gradient ascent step. Meanwhile, the gradient $\lambda$ taken is an approximation of the gradient of $d(\lambda)$. Thus, by subtracting $d(\lambda)$, we can obtain that $\Phi$ will decrease during iterations. Therefore, the first thing is to show the lower bound of the function $\Phi$.

**Lemma 1** (Lower bound of $\Phi$). *Suppose* **(A1)-(A4)** *hold. It holds that $\Phi_x(w, \lambda)$ is bounded below by $\underline{f}$.*

The proof of this lemma is basically due to the definition of $\Phi_x(w, \lambda)$ and $d(\lambda)$. Then, similar to the proof of gradient descent, we give a lemma that shows the descent of potential function under certain choices of hyperparameters.

**Lemma 2** (Potential function descent, proof can be found in Lemma **??** in Appendix). *Suppose* **(A1)-(A6)** *hold. In addition, we assume $\Gamma > \frac{D_\lambda L_2 + L_1}{\mu}$, it holds that*

$$\mathbb{E}[\Phi_x(w_{t,k}, \lambda_{t,k}) - \Phi_x(w_{t,k+1}, \lambda_{t,k+1})] \leq -C_1\mathbb{E}\|\nabla_w L_x(w_{t,k}, \lambda_{t,k})\|^2 - C_2\mathbb{E}[\|\lambda_t - \lambda_t^*\|^2] + C_3\sigma^2,$$

*where $\lambda_t^+ = \Pi_\Lambda(\lambda_t + \eta_\lambda \nabla d(\lambda_t))$, $C_1 = \Theta(\eta_w - \eta_w^2 - \eta_\lambda^2 - \eta_\lambda)$, $C_2 = \Theta(\eta_\lambda)$ and $C_3 = O(\eta_w^2 + \eta_\lambda^2)$*

Thus, when choosing sufficient small $\eta_w$ and $\eta_\lambda$, we can achieve positive $C_1$ and $C_2$. Together with the lower bound of the function $\Phi$, the convergence of the inner algorithm can be shown. Because of the uniqueness of the KKT point, by choosing $\eta_w$ and $\eta_\lambda$ in order of $1/\sqrt{K}$, it can be shown that

$$\frac{1}{K}\sum_{k=1}^{K}\mathbb{E}\|w_{t,k} - w^*(x_t)\|^2 = O(1/\sqrt{K}), \quad \frac{1}{K}\sum_{k=1}^{K}\mathbb{E}\|\lambda_{t,k} - w^*(x_t)\|^2 = O(1/\sqrt{K}).$$

Therefore, with the convergence rate of $w_{t,k}$ and $\lambda_{t,k}$ and equation 4, we can easily prove theorem 1.

### 4.2 Proof Sketch of Theorem 2

To apply stochastic gradient descent analysis on $x$, although we have smoothness for the function $f_0, f_1, \cdots, f_N$ on $w$, we need to verify the smoothness of $f_0(w^*(x))$ with respect to $x$.

**Lemma 3** (Convergence of stochastic gradient descent with biased gradient estimation, proof can be found in Lemma **??** in Appendix). *Suppose function $f(x)$ is lower bounded by $\underline{f}$ with $L$-Lipshitz gradient. $g(x)$ is an unbiased gradient estimator of $\nabla f(x)$ satisfying that expected norm of $g(x)$ are bounded by $G$ in domain $\mathcal{X}$ for function $f$. Then with update rule $x_{t+1} = \Pi_\mathcal{X}(x_t - \eta_x(g(x_t) + \xi_t))$, where $\eta_x = \Theta(1/\sqrt{T})$, $\mathcal{X}$ is a convex set and $\mathbb{E}\|\xi_t\|^2 \leq \epsilon^2$. By defining $\hat{x} = \arg\min_{y\in\mathcal{X}}(f(y) + \frac{\rho}{2}\|y - x\|^2)$ and $\bar{\nabla}_\rho f(x) = \rho(x - \hat{x})$, where $\rho = 2L$, then it holds that*

$$\frac{1}{T}\mathbb{E}\sum_{t=1}^{T}\|\bar{\nabla}_\rho f(x_t)\|^2 = O(1/\sqrt{T} + \epsilon^2).$$

As Lemma 3 suggests, when $f_0(w^*(x))$ satisfying $L$-Lipschitz gradient, bounded estimation error, and bounded gradient norm, the convergence rate can achieve $O(1/\sqrt{T})$ with an error term related to estimation error. Theorem 1 shows the estimation error can be bounded by $O(1/\sqrt{K})$. Combining these two results we can prove Theorem 2.

# 5   Experimental Results

In this section, we compare our algorithm with other bi-level optimization algorithms (FEDNEST (Tarzanagh et al., 2022), SUSTAIN (Khanduri et al., 2021) and RFHO (Franceschi et al., 2017)) in two cases: the toy example and two vision tasks. Further, in vision tasks, agnostic federated learning (AFL) is tested (Mohri et al., 2019). When k local steps are used in each algorithm, FEDNEST, RFHO, and our algorithm will perform $2kd$ real number transmission, where $d$ is the dimension of optimization. SUSTAIN will perform $(k + 1)d$ real number transmission. In the vision tasks, they perform the same real number of transmissions as $k = 1$. For a fair comparison, we set the local device update to 1.

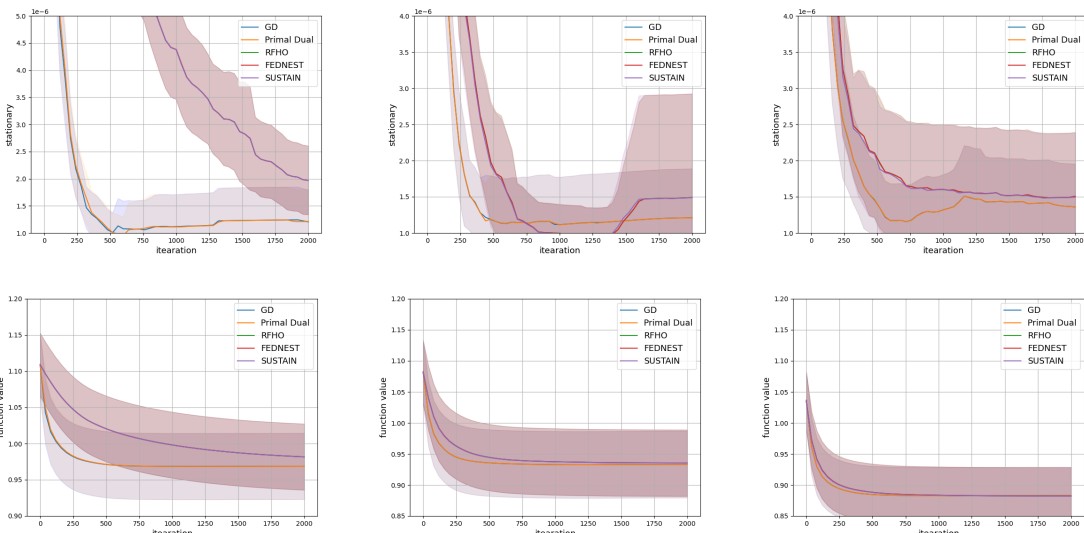

Figure 1: The figure shows the result of the toy example where all clients participate in the optimization process in each iteration, and all gradient and hessian are estimated without noise. The above line shows the stationary of $x$ in each iteration, and the second row shows the function value of $x$ ($f(w^*(x))$). The left column shows the results when the number of local steps is 1; the middle column shows the results of 5 local steps, and the right column gives the results of 10 local steps. The shadow part of the function value corresponds to the 0.1 standard error area, and the shadow part in stationary corresponds to the 0.5 standard error area.

## 5.1   Toy Example

In this section, we apply algorithms to solve problem 1 with $f_i$ as follows:

$$f_i(w) = \frac{1}{2}\|A_i w - B_i\|^2 + cos(a_i^\top w - b_i),$$

where $A_i \in \mathbb{R}^{30 \times 20}, B_i \in \mathbb{R}^{30}, a_i \in \mathbb{R}^{20}$ and $b_i \in \mathbb{R}$ are all generated from Gaussian distribution. The variance of each component in $A_i$ and $a_i$ is $1/\sqrt{20}$ and the variance of each component in $B_i$ is $1/\sqrt{30}$ and variance of $b_i$ is 1. When generated function $f_i$ is not 0.1-strongly convex, we randomly generate a new one until we get strongly convex $f_i$ whose modular is not less than 0.1. Three local steps ($K$=1, 5, 10) are tested. Here, the local steps are used for $w$ update for algorithm FEDNEST, RFHO, and our algorithm, and the local steps are used for Hessian estimation for algorithm FEDNEST and SUSTAIN. Because we can easily compute the Hessian matrix and its inverse for this toy example, we test the algorithm using the inverse of the estimated Hessian to compute the gradient of $x$ named GD. We test two settings of the toy example. One is the deterministic setting, where no estimation noise or client disconnection will occur. In the other setting, we add white Gaussian noise with a noise level of 0.5 in each estimation (including gradient estimation and Hessian estimation). Also, each client has a 0.5 probability of connecting with the server.

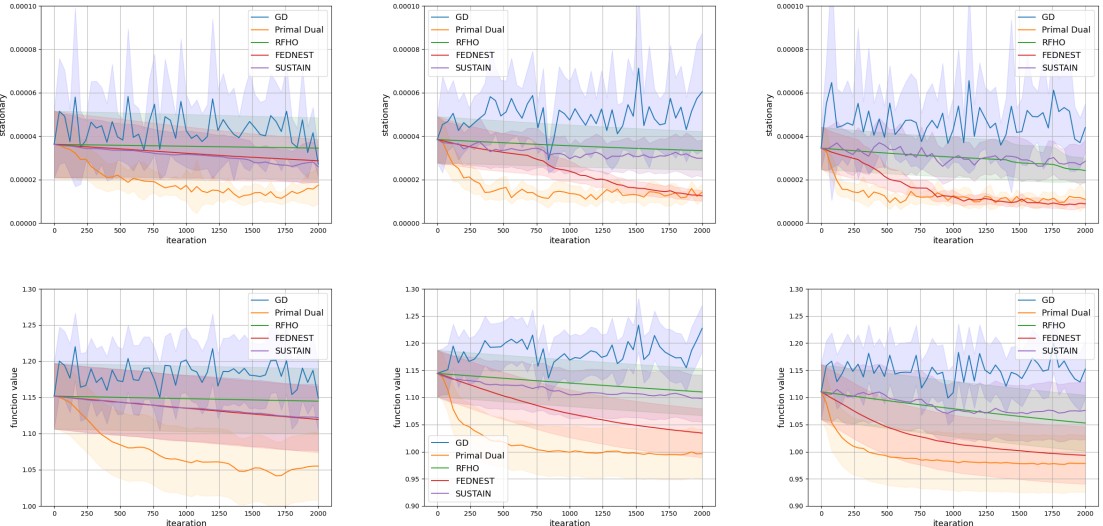

Figure 2: The figure shows the result of the toy example where the active rate is 0.5 in each iteration, and all gradient and hessian are estimated with white-Gaussian noise with a noise level of 0.5. The above line shows the stationary of $x$ in each iteration, and the second row shows the function value of $x$ ($f(w^*(x))$). The left column shows the results when the number of local steps is 1; the middle column shows the results of 5 local steps, and the right column gives the results of 10 local steps. The shadow part of the function value corresponds to the 0.1 standard error area, and the shadow part in stationary corresponds to the 0.5 standard error area.

To evaluate the performance of different algorithms, we calculate the function value of $f_0(w^*(x))$ and the stationary of x, i.e. $x - \Pi_X(x - 0.001\nabla_x f_0(w^*(x)))$, where $w^*(x)$ is approximate by 200 gradient steps. We take 15 local clients and run 20 times and get the results of different algorithms. The results of the deterministic setting are shown in Figure 1, and the results of the noise setting are shown in Figure 2.

As it is shown in Figure 1, with local steps getting larger and larger, the performance of FEDNEST, RFHO, and SUSTAIN is getting close to GD, while the performance of the primal-dual method is similar to GD whatever local step used in the algorithm even with only one single step. When noise is added in the Hessian, the direct inverse may cause the biased estimation. Thus, the performance of GD gets much worse than it in the deterministic setting shown in Figure 2. Also, in Figure 2, our algorithm can perform better than other algorithms when the local step is small. When local steps increase to 10, FEDNEST and our algorithm have competitive results.

### 5.2 Vision Tasks

In this section, we apply algorithms to train LeNet5(LeCun et al., 1998) on dataset MNIST(LeCun et al., 1998) and Fashion-MNIST(Xiao et al., 2017). To construct non-iid datasets on different local clients and the global server's validation set, we randomly pick 20 samples per label out of the whole training dataset and form the validation set. Then, the rest of the training data are divided into 3 sets, and each set will be assigned to a local client. The first client contains samples labeled as 0,1,2,3,4, the second client contains samples labeled as 5,6,7, and the third client contains samples labeled as 8,9 for all two datasets. To test the algorithm's ability to choose the proper coefficient of local clients, we add 7 noise nodes containing 5000 samples with random labels. We set the learning rate of $w$ to be a constant learning rate without any decay selected from $\{0.1, 0.01, 0.001\}$ for all training methods, and the learning rate of $x$ is selected from $\{0.1, 0.01, 0.001, 0.0001\}$. The batch size for all three training cases is set to 64. $\Gamma$ used in the proposed algorithm is selected from $\{0.5, 1, 2\}$. For simplicity, we set the local step as 1. We run 2000 iterations for MNIST and 6000 iterations for Fashion-MNIST. Active probability is set in $\{0.5, 0.9, 1\}$. We compare the

Table 1: Test Accuracy and $x$ output of Training LeNet 5 on MNIST. "AP" represents Active Probability, and Accuracy stands for Test Accuracy.

| AP | | RFHO | FEDNEST | SUSTAIN | Ours |
|---|---|---|---|---|---|
| 1 | Accuracy | $98.34\% \pm 0.18\%$ | $98.15\% \pm 0.23\%$ | $99.02\% \pm 0.15\%$ | $98.43\% \pm 0.17\%$ |
| | $x^{(1)}$ | $0.488 \pm 0.104$ | $0.425 \pm 0.081$ | $0.411 \pm 0.069$ | $0.455 \pm 0.016$ |
| | $x^{(2)}$ | $0.311 \pm 0.104$ | $0.245 \pm 0.133$ | $0.305 \pm 0.045$ | $0.334 \pm 0.020$ |
| | $x^{(3)}$ | $0.197 \pm 0.031$ | $0.294 \pm 0.176$ | $0.282 \pm 0.029$ | $0.212 \pm 0.026$ |
| | $x^{(4),\cdots,(10)}$ | $\sim 6e - 4$ | $\sim 6e - 3$ | $\sim 3e - 4$ | $\sim 2e - 4$ |
| 0.9 | Accuracy | $98.07\% \pm 0.4\%$ | $98.09\% \pm 0.21\%$ | $98.85\% \pm 0.29\%$ | $98.43\% \pm 0.19\%$ |
| | $x^{(1)}$ | $0.407 \pm 0.040$ | $0.395 \pm 0.136$ | $0.386 \pm 0.058$ | $0.449 \pm 0.046$ |
| | $x^{(2)}$ | $0.281 \pm 0.065$ | $0.314 \pm 0.045$ | $0.345 \pm 0.028$ | $0.333 \pm 0.050$ |
| | $x^{(3)}$ | $0.291 \pm 0.018$ | $0.239 \pm 0.085$ | $0.265 \pm 0.038$ | $0.217 \pm 0.024$ |
| | $x^{(4),\cdots,(10)}$ | $\sim 4e - 3$ | $\sim 8e - 3$ | $\sim 7e - 4$ | $\sim 2e - 4$ |
| 0.5 | Accuracy | $97.86\% \pm 0.36\%$ | $95.37\% \pm 4.10\%$ | $97.60\% \pm 0.49\%$ | $98.24\% \pm 0.23\%$ |
| | $x^{(1)}$ | $0.449 \pm 0.090$ | $0.539 \pm 0.076$ | $0.365 \pm 0.015$ | $0.468 \pm 0.052$ |
| | $x^{(2)}$ | $0.276 \pm 0.075$ | $0.217 \pm 0.059$ | $0.329 \pm 0.013$ | $0.372 \pm 0.053$ |
| | $x^{(3)}$ | $0.271 \pm 0.129$ | $0.210 \pm 0.039$ | $0.292 \pm 0.015$ | $0.16 \pm 0.035$ |
| | $x^{(4),\cdots,(10)}$ | $\sim 6e - 4$ | $\sim 6e - 3$ | $\sim 2e - 3$ | $\sim 2e - 4$ |

test accuracy among different methods. As a baseline, we report the test accuracy for training with the validation set only named val, training with the average loss of each client named avg, and training with $x = (0.5, 0.3, 0.2, 0, \cdots, 0)$ named opt. All experiments run on V100 with Pytorch (Paszke et al., 2019). Results are shown in Figure 3, Figure 4, and Table 1.

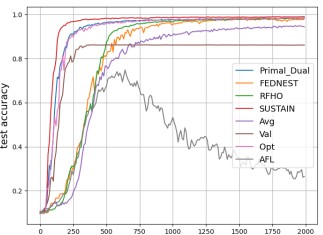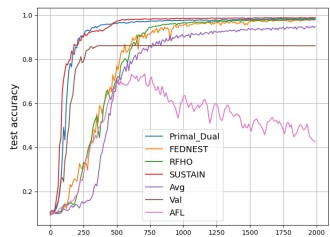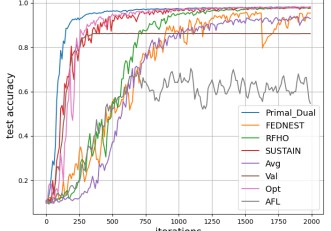

Figure 3: Test accuracy of training LeNet 5 on MNIST dataset. The left curve shows the result when the active probability is 1; the middle curve shows the result when the active probability is 0.9, and the right curve shows the result with the active probability of 0.5.

Figure 3 shows the test accuracy of the MNIST dataset with different active probabilities. Although SUSTAIN works better than the primal-dual algorithm when all local devices participate in the optimization process, when clients' participant rate decreases to 0.5, SUSTAIN works worse than our method. Primal-dual becomes slower than SUSTAIN may be because of the initialization of the dual variable. When the dual variable is far from its real value it needs more time to get a good enough point. Other than SUSTAIN, our algorithm can converge faster and more stable to a high accuracy point. Further, we list the output of $x$ and standard error of test accuracy for 5 different experiments for different algorithms in Table 1. According to Table 1, our algorithm can achieve a more stable output with respect to $x$, and the output $x$ is closer to $0.5, 0.3, 0.2$, which is related to the number of labels the first three clients holds.

Figure 4 gives the test accuracy of training LeNet 5 on the Fashion-MNIST Dataset. Similar to the results of the MNIST dataset, when the clients' participant is high (0.9,1), SUSTAIN works slightly better than the primal-dual algorithm. But when more local devices disconnect to the server, the performance of SUSTAIN drops, while the primal-dual algorithm remains fast convergence speed and high test accuracy.

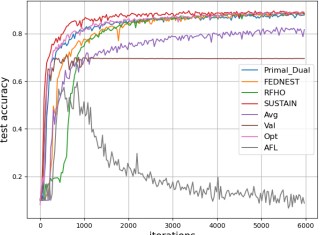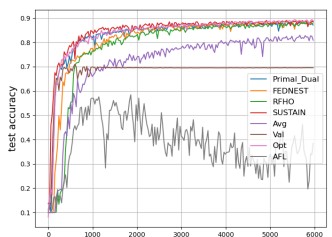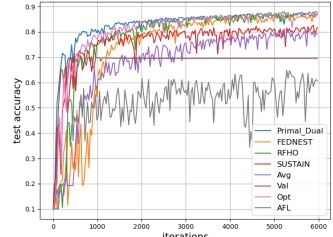

Figure 4: Test accuracy of training LeNet 5 on the Fashion-MNIST dataset. The left curve shows the result when the active probability is 1; the middle curve shows the result when the active probability is 0.9, and the right curve shows the result with 0.5 active probability.

## 6    Conclusion

In this paper, we proposed a primal-dual-based method for solving a bi-level optimization problem based on a federated learning task (local coefficient learning). We give a theoretical analysis that shows the convergence of the proposed algorithm. Though the analysis shows it needs more iterations for the algorithm to converge to an $\epsilon$-stationary point, it works well with a pretty small number of local steps in both toy case and neural network training. Other than that convergence rate can be improved (perhaps it should be in the order of $O(1/\sqrt{T})$ instead of $O(1/\sqrt{T} + 1/\sqrt{K})$), the initialization of dual variable affects the speed for convergence, which we leave as the future work.

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
