# OpenReview forum: "Efficient Weighting and Optimization in Federated Learning: A Primal-Dual Approach"
_TMLR — Rejected by TMLR_

### Review · Reviewer_kEM3 · 2023-07-14

**Summary Of Contributions:**

The paper considers the bi-level optimization problem ((1) in the paper) in the context where the functions $f_i$ are distributed between nodes. The authors propose a new algorithm that solves the initial problem without calculating the inverse of Hessians using the primal-dual optimization technique. The algorithm is supported by proven theoretical convergence rates and experiments.

**Audience:**

Yes

**Broader Impact Concerns:**

.

**Claims And Evidence:**

Yes

**Requested Changes:**

See "Weaknesses" and "Questions" in "Strengths And Weaknesses."

**Strengths And Weaknesses:**

The initial problem is important for the community, though I didn't understand some details (see my comments further). The proposed method is interesting. I didn't check the proof in detail, though I went through it and checked some parts of it. Intuitively, I can believe that we can find the estimators of gradients for a fixed $x$ using the primal-dual approach.

However, there is at least on major weakness that I want the author to clarify:

~~Major weakness:~~

~~The initial problem (1) has the nonconvex constraint $X = [(x_1, \dots, x_N) | x_i \geq 0, |x|_1 = 1].$ In Algorithm 2, the point $ x^{t+1} $ is defined as a projection on the nonconvex set $X$. However, in Lemma 3/Lemma 14, the authors assume $X$ is a convex set. So Lemma 3/Lemma 14 can not be applied. This is a major flaw.~~

~~Even if we can inverse Hessians and use Kolstad & Lasdon (1990)'s result, it is not clear if the projected gradient descent method will converge to the minimum with the nonconvex constraint $X.$~~

(crossed the weakness because it was resolved in the following comments)

Other weaknesses:

**Weaknesses:**
1. In the context of FL, it was never defined what $f_0$, $f_i$ mean. In practice, how do people choose $f_0$? Even in the experiments, $f_0$ is not defined...
2. Please fix all typos. For instance, "Proof of Theorem 2" is full of typos. There is a problem with the balance of brackets in the proof of Lemma 14...
3. I could not find the proofs of the propositions. Can you at least add relevant references?
4. For theoreticians, the constants from assumptions are very important. Can you unhide the constants in Theorem 1 and 2? It will be very difficult for future works to compare with Theorem 1 and 2.

Questions:
1. Can you kindly explain how the additional term $+ \Gamma \sum_{i=1}^N x^i f_i(w)$ helps in the theory? In page 4, you say that "Thus, the problem 2 becomes (5)." It is not clear because (5) depends on $\Gamma$ while (2) does not depend on $\Gamma.$ Why the optimal $w$ in (5) do not depend on $\Gamma$?

---

> ### Author Response · Authors · 2023-07-26
> **To Reviwer kEM3(1/2)**
>
> Thank you for giving constructive suggestions. we will respond to each issue in the following.
>
> > ***The initial problem (1) has the nonconvex constraint.***
> >
> > > The constraint in problem (1) is convex. When we restrict x in the domain $ x_i \geq 0 $, the constraint  $||x||_1 = 1$
> > >
> > > is equivalent to
> > >  $\sum_{i=1}^N x_i=1$, which is a convex constraint. Meanwhile, $x_i \geq 0 $ are convex constraints.
>
> > ***In the context of FL, it was never defined what $f_0$ , $f_1$  mean. In practice, how do people choose? Even in the experiments, $f_0$ is not defined...***
> >
> > > As we introduce a validation set in the global server, we define $f_0$ as the validation loss in the global validation set. We define $f_i$ as the training loss in each local client with their own training data.
> > >
> > > To be more specific in the experiment, $f_0$ is the objective evaluated on 200 samples selected in the global server, and $f_i$ is the objective function evaluated on data samples in the ith device.
>
>
> > ***Please fix all typos.***
> >
> > > Thanks for pointing out typos, we correct the typos and check all the proof again.
>
>
> > ***I could not find the proofs of the propositions. Can you at least add relevant references? ***
> >
> > > Sorry about the missing proofs of the propositions. We have added the proof of each proposition in the Appendix. We list the proof in the following:
>
> ***Proof of Proposition 1***
> > By the optimality condition of $w^*(x)$, it holds that
> \begin{aligned}
> \nabla_w \sum_{i=1}^{N}x^{(i)} f_i(w^*(x)) = 0.
> \end{aligned
> >
> > Then, taking the gradient of $x^{(k)}$ for both sides of the above equation, it holds that
> \\[
> \begin{aligned}
> 0 &= \nabla_{x^{(k)}} 0\\\\
> &= \nabla_{x^{(k)}} \nabla_w \sum_{i=1}^N x^{(i)} f_i(w^*(x))\\\\
> & = \nabla_w f_k(w^*(x)) + \sum_{i=1}^N x^{(i)} \nabla^2_w f_i(w^*(x))\frac{\partial w^*(x))}{\partial x^{(k)}}\\\\
> \end{aligned}
> \\]
> >
> > Meanwhile, because $f_i(w)$ is a strongly convex function, the Hessian of $f_i(w)$ (i.e., $\nabla_w^2 f_i(w)$) is nonsingular. Together with $x^{(i)} \geq 0$ and $\sum_{i=1}^N x^{(i)} = 1$, $\sum_{i=1}^N x^{(i)} \nabla_w^2 f_i(w)$ is nonsingular. Thus, by arranging terms, it holds that
> \\[
> \frac{\partial w^*(x)}{\partial x^{(k)}} = - \left(\sum_{i=1}^N x^{(i)} \nabla_w^2 f_i(w^*(x))\right)^{-1} \nabla_w f_k(w^*(x))
> \\]
> >
> > By the chain rules it holds that
> \\[
> \frac{\partial f_0(w^*(x))}{\partial x^{(k)}} = \nabla_w f_0(w^*(x))^\top {\frac{\partial w^*(x)}{\partial x^{(k)}}}  = - \nabla_w f_0(w^*(x))^\top   \left(\sum_{i=1}^N x^{(i)} \nabla_w^2 f_i(w^*(x))\right)^{-1} \nabla_w f_k(w^*(x)).
> \\]
>
> ***Proof of Proposition 2***
> > Because $x_i \geq 0$ and $||x||_1 = 1$,
> >
> > $\sum_{i=1}^N x^{(i)} f_i(w)$  is a strongly convex function.
> >
> > Thus, $\sum_{i=1}^N x^{(i)} \nabla_w^2 f_i(w)$ is nonsingluar, which means each row of $\nabla_w \sum_{i=1}^N x^{(i)}\nabla_w f_i(w)$ is linear independent for all $w$.
> >
> > Therefore, the constraint satisfies Linear Independent Constraint Qualification.
> >
> > As the Lagrange function can be rewritten as $f_0(w) + \lambda^\top \sum_{i=1}^N x^{(i)}\nabla_wf_i(w)$, we can give the KKT condition by taking gradient with respect to primal variables and dual variables, which gives:
> \\[
> \begin{aligned}
>  &   \nabla_w f_0(w) + \sum_{i=1}^N x^{(i)}\nabla_w^2f_i(w)\lambda = 0,\\\\
>  &   \sum_{i=1}^N x^{(i)}\nabla_w f_i(w) = 0.
> \end{aligned}
> \\]
>
> ***Proof of Proposition 3***
> > The stationary point of $\min_w\max_\lambda L_x(w,\lambda)$ satisfies the following condition:
> \\[
> \begin{aligned}
>  &   \nabla_w f_0(w) + \sum_{i=1}^N x^{(i)}\nabla_w^2f_i(w)\lambda + \Gamma \sum_{i=1}^N x^{(i)} \nabla_w f_i(w)= 0,\\\\
>  &   \sum_{i=1}^N x^{(i)}\nabla_w f_i(w) = 0.
> \end{aligned}
> \\]
> >
> > Because $ \sum_{i=1}^N x^{(i)}\nabla_w f_i(w)$ is a strongly convex function, the optimal solution is unique. We denote the optimal solution as $w^*(x)$.
> >
> > Thus, we can calculate the $\lambda^*(x)$ based on the first equation $\lambda^*(x) = -\left(\sum_{i=1}^N x^{(i)}\nabla_w^2f_i(w^*(x)) \right)^\top \nabla_w f_0(w^*(x))$.
> >
> > Then, plugging $w^*(x)$ and $\lambda^*(x)$, we can find that $w^*(x)$ and $\lambda^*(x)$ satisfy KKT condition.
>
> ***Proof of Proposition 4***
> > Let $\lambda^*(x)$ be the optimal solution without constraint.
> >
> > By the definition of $\Lambda$, it holds that $\lambda^*(x) \in \Lambda$.
> >
> > Thus, with the optimality condition, $\lambda^*(x)$ will be the only solution to problem (5).
> >
> > Therefore, the solution to the problem (5) is the solution to the problem (2).

---

> > ### Comment · Reviewer_kEM3 · 2023-07-26
> > **Regarding "The initial problem (1) has the nonconvex constraint."**
> >
> > Indeed, I somehow overlooked this simple fact. Thank you.
> > I will retract the "Major Weaness" issue and change "Claims And Evidence" to "Yes".
> >
> > I will try to respond to other comments later.

---

> > ### Comment · Reviewer_kEM3 · 2023-08-09
> > **Respond**
> >
> > Thank you!
> >
> > All my comments and weaknesses were successfully addressed except for the last one:
> > > For theoreticians, the constants from assumptions are very important. Can you unhide the constants in Theorem 1 and 2? It will be very difficult for future works to compare with Theorem 1 and 2.
> >
> > I will recommend accepting this paper if the authors either add all hidden constants or give strong evidence why they are omitted.

---

### Review · Reviewer_Hz3F · 2023-07-19

**Summary Of Contributions:**

This submission aims to applying weights to clients in federated learning. The weights are trained to optimize the performance on a validation set on the server. The authors formulate this problem into a bilevel problem with a simplex constraint. Algorithms are proposed to optimize the target problem.

**Audience:**

Yes

**Broader Impact Concerns:**

No.

**Claims And Evidence:**

Yes

**Requested Changes:**

See above.

**Strengths And Weaknesses:**

Strengths: Analysis and experiments are given to support the problem formulation and algorithms. Analysis has been done for a couple of different cases to fit with different problems.

Weaknesses:

1) In terms of the contribution on the problem formulation, the following paper has considered a very similar, if not identical, bilevel problem.

Huang, Yankun, et al. "Federated learning on adaptively weighted nodes by bilevel optimization." arXiv preprint arXiv:2207.10751 (2022).

They also apply weights to the clients by a bilevel problem with constraints.

2) Note that in the the above paper the complexity looks much better than this submission (e.g. Theorem 3). Can the authors refer to that paper and discuss the difference and relation between this submission and the above paper, in terms of algorithm and analysis? Also in terms of experiments if possible.

3) Looks that in Algorithm 2, communication is required in every iteration of the inner loop, therefore the commutation complexity should be $O(TK)$? If so, then the complexity is even higher.

---

### Review · Reviewer_5BZV · 2023-08-05

**Summary Of Contributions:**

The paper formulates the problem of finding optimal weighting of local loss function in federated learning as a bi-level optimization problem. The paper proposes a Lagrangian-based approach to solve the bi-level formulation and establishes convergence guarantee. Numerical experiments on synthetic and real data sets are provided to illustrate the performance of the proposed method.

**Audience:**

Yes

**Broader Impact Concerns:**

I do not have concerns related to this criteria.

**Claims And Evidence:**

Yes

**Requested Changes:**

I have the following questions/suggestions to improve the paper:

- Do the authors claim the novelty in the design of the Langrange function (3) or it has been used in previous works. If it is the former, the authors can further emphasize this in the paper. Otherwise, adding relevant papers is recommeded.
- ~~Page 17, I think there should be $L_1$ in $\frac{1}{2\eta_\lambda} - \frac{1}{\eta_\lambda}$. This affects all following calculation so I suggest the authors revise this.~~ Update: after authors response, I am clear on this point so crossing this out.

Minor comments:

- After Theorem 1 in main paper, it is unclear what "the gradient of x" is. It should be the gradient of $f_0$ w.r.t. $x$.
- It may be better to highlight the best metric per row in Table 1.
- In experiments, I find it hard to know how many local clients are there. In the second experiment, there are 10 local clients which is rather small in federated learning setting where the number of local clients is large.
- $\Gamma$ appears to be one of important parameter to choose in the proposed algorithm but I do not find much discussion on choosing this apart from setting it to 1 in the experiments. Theoretically, we need the knowledge of $D_\lambda$ (or $D_w$) which is hard to get in practice.
- Proofs have numerous typos, some that I can spot are: many in equation (10), second equation in the proof of Lemma 11 with redundant brackets, page 21 with missing right bracket and $\xi_t^(i) \to \xi_t^{(i)}$.
- Beginning of page 17, I think $L(w_t,\lambda_t)$ can be greater or equal to $d(\lambda_t)$ not strictly greater.

**Strengths And Weaknesses:**

**Strengths**:

- The paper focuses on finding optimal weighting of the local loss function which should be better than then conventional uniform weighting. The bi-level optimization problem is more challenging to solve.
- The proposed method has theoretical convergence guarantee. I have checked the proofs, most steps are fine but I do have some questions to make sure the proofs are correct.

**Weaknesses**:

- The proofs contain numerous typos, some of them are major which affect the final results.
- In federated learning, there is also a metric of interest which is the communication cost (amount of data communicated between server and clients). The paper does not discuss about the communication complexity/cost between the proposed method and existing ones.

---

> ### Author Response · Authors · 2023-08-18
> **To Reviewer 5BZV**
>
> Thank you for giving constructive suggestions. we will respond to each issue in the following.
>
> ### Do the authors claim the novelty in the design of the Langrange function (3) or it has been used in previous works. If it is the former, the authors can further emphasize this in the paper. Otherwise, adding relevant papers is recommeded.
>
> > We design the Lagrange function (3) on our own. We have stated the contribution in the third bullet of the contributions where we claim that we give a new augmented term.
>
> ### Page 17, I think there should be $L_1$ in $\frac{1}{2\eta_\lambda} - \frac{1}{\eta_\lambda}$. This affects all following calculation so I suggest the authors revise this.
>
> > The inequality is correct, since
> \begin{aligned}
> &2L_1||\lambda_{t+1} -\lambda_t||(||w_t - w^*(\lambda_t)|| + ||w_{t+1} - w_t||) \\\\
> &= \frac{1}{\sqrt{\eta_\lambda}} ||\lambda_{t+1} -\lambda_t||(2L_1\sqrt{\eta_\lambda} ||w_t - w^*(\lambda_t)|| + 2L_1\sqrt{\eta_\lambda}||w_{t+1} - w_t||) \\\\
> &\leq \frac{1}{2\eta_\lambda} \|\lambda_{t+1} -\lambda_t||^2 + 2L_1^2\eta_\lambda \|w_t - w^*(\lambda_t)|| + 2L_1^2\eta_\lambda||w_{t+1} - w_t||.\end{aligned}
> > Thus, there is no $L_1$ in the term $\frac{1}{2\eta_\lambda} - \frac{1}{\eta_\lambda}$
>
> ### After Theorem 1 in main paper, it is unclear what "the gradient of x" is. It should be the gradient of $f_0$ w.r.t $x$.
> > Thank you for the comment, we have updated the manuscript.
>
> ### It may be better to highlight the best metric per row in Table 1.
> > Thanks for the suggestion, we add the underline under the best result for each row.
>
> ### In experiments, I find it hard to know how many local clients are there. In the second experiment, there are 10 local clients which is rather small in federated learning setting where the number of local clients is large.
>
> > We use 15 local clients in the toy experiments. We are sorry that due to the computational issue, we can not provide the results of more than 10 clients in the current version.
>
> ### $\Gamma$ appears to be one of important parameter to choose in the proposed algorithm but I do not find much discussion on choosing this apart from setting it to 1 in the experiments. Theoretically, we need the knowledge of $D_\lambda$ (or $D_w$) which is hard to get in practice.
>
> > We test on the case where $\Gamma = 0.5,2$. Here is the results.
>
> | Gamma\AP | 1     | 0.9   | 0.5   |
> |----------|-------|-------|-------|
> | 0.5      | 98.17 | 96.91 | 96.89 |
> | 1        | 98.43 | 98.43 | 98.24 |
> | 2        | 98.35 | 98.35 | 98.00 |
> > Similarly to the learning rate whose selection is related to the unknown Lipschitz constant, we have to tune $\Gamma$. Since we test when $\Gamma = 1$, the algorithm performs well, we do not test more and just set $\Gamma = 1$. We have already change the text to $\Gamma$ is selected from $\{0.5,1,2\}$
>
>
> ### Proofs have numerous typos, some that I can spot are: many in equation (10), second equation in the proof of Lemma 11 with redundant brackets, page 21 with missing right bracket and $\xi_t^(i) \rightarrow \xi_t^{(i)}$.
>
> >Sorry for the typos, we have revised all brackets in the appendix.
>
> ### Beginning of page 17, I think $L(w_t,\lambda_t)$ can be greater or equal to $d(\lambda_t)$ not strictly greater.
>
> > Sorry for the typo, we have fixed it in the manuscript.

---

> > ### Comment · Reviewer_5BZV · 2023-09-17
> > **Re: author responses**
> >
> > I thank the authors for providing responses to my comments. Now I see inequality on page 17 is in fact correct. I have no more concerns about the current version of the paper.

---

### Decision · Action_Editors · 2023-10-17

**Recommendation:** Reject

**Comment:**

Two reviewers endorsed acceptance at the initial recommendation stage, while one expressed reservations about the paper. During discussions, a consensus emerged regarding the presence of the bilevel programming formulation in Huang et al.'s work (2022). Additionally, the authors did not adequately address the reviewers' recurring concerns about the high communication complexity ($O(\epsilon^{-8})$) associated with the proposed algorithm.

Considering these factors, I would **NOT recommend accepting the current version and instead encourage a resubmission**.

In the resubmitted version, I recommend that the authors conduct a **thorough comparative analysis of their algorithms with the relevant references** highlighted by the referees, such as Huang et al. (2020) and Prashant Khanduri et al. (2023). Furthermore, it is crucial for the authors to provide a **clear, compelling response to the concerns raised regarding the high communication complexity** of the proposed methods.

**Audience:**

Federated learning will definitely interest the TMLR's audience.

**Claims And Evidence:**

The paper claims that one main contribution is to formulate the local coefficient weights by a bilevel problem. However, as one reviewer pointed out this formulation was done in the paper Huang et al. (2022).  The communication complexity in this submission seems to be worse than the existing work.

**Resubmission Of Major Revision:**

The authors may consider submitting a major revision at a later time.